

# The potential of GNSS radio occultation data for the analysis of the tropical width: a comparison with reanalyses

Annika Reiter[1], Julia Danzer[1], Andrea K. Steiner[1]

[1]Wegener Center for Climate and Global Change, University of Graz, Graz, 8010, Austria

*Correspondence to*: Julia Danzer (julia.danzer@uni-graz.at)

**Abstract.**

The tropics are expanding poleward as a result of anthropogenic climate change. This in turn has great implications on the temperature and precipitation patterns in the subtropical regions. Previous studies have found varying widening trends, most of which have been derived using reanalysis and climate model data. These trend discrepancies underline the need for studies

using alternative datasets. Here, we explore the potential of GNSS radio occultation (RO) data for analyzing the tropical width as an independent observational source of information with key characteristics: high accuracy, global availability, and long-term consistency. We evaluate the skill of RO temperature and newly established RO wind records to accurately capture tropical width features, using tropopause break and jet stream metrics. The results are compared to three state-of-the-art reanalysis datasets (i.e., ERA5, MERRA-2, and JRA-3Q). Zonal-mean patterns and the regional structure of tropical width

features are investigated to test the utility of RO in respect to its spatial robustness. Furthermore, we provide a perspective on the necessary record length for reliable trend estimation of the tropical width. Comparisons of RO to reanalyses show overall high agreement of the zonal-mean values. As for the zonally resolved metrics, results from reanalyses and RO align well with exceptions over the northern hemisphere. While the RO record length is still a bit too short for detecting tropical width trends, the results are encouraging and confirm that RO is a valuable alternative observation-based dataset, with increasing relevance

towards the future.

## 1 Introduction

Commonly the tropical edge in response to climate change is investigated using climate model output (e.g., Chemke and Polvani, 2019; Waugh et al., 2018) and reanalysis records (e.g., Davis and Davis, 2018; Nguyen et al., 2018; Staten et al., 2018). Most of these studies reveal that the tropics are expanding poleward due to anthropogenic climate change (e.g., Davis

and Birner, 2013; Grise et al., 2019; Hu et al., 2018; Staten et al., 2019), which has great implications on the temperature and precipitation patterns in the subtropics (Amaya et al., 2018; Feng and Fu, 2013; Sharmila and Walsh, 2018; Xian et al., 2021). However, the reported trend results of the tropical width are not fully consistent (Staten et al., 2018) and often overlook the complex uncertainty of the underlying dataset (Baldassare et al., 2023). In the case of reanalyses, the error characteristics include, the amount and source of assimilated data combined with model uncertainties, such as assigned weights or set



parameters (Hoffman et al., 2017; Long et al., 2017; Parker, 2016). This in turn, can lead to biases in the trend estimation and thus merits comparison and validation with observations.

Global navigation satellite systems (GNSS) radio occultation (RO) provides atmospheric information, with clearly quantified error characteristics, including the sampling error and structural uncertainty (Scherllin-Pirscher et al., 2011; Steiner et al., 2013) – posing an alternative, observation-based dataset for studying tropical width metrics. The GNSS-RO technique has the

decisive advantage of long-term stability with no need for inter-calibration between satellite missions because it is based on precise time measurements. Data are provided with global coverage, near-all weather capability, and a specifically high-vertical resolution in the upper troposphere and lower stratosphere (UTLS) (e.g., Angerer et al., 2017; Foelsche et al., 2011; Scherllin-Pirscher et al., 2021; Steiner et al., 2020a). In the low- to mid-latitudes of the upper troposphere a resolution of about 100 m to 200 m, and about 500 m in the lower stratosphere can be attained (Zeng et al., 2019). This is particularly interesting

for the investigation of features in the troposphere and the tropopause region, such as variability and changes of the tropical width. Furthermore, the location of the measured air parcel can be precisely positioned, which enables independent information on altitude and pressure (Scherllin-Pirscher et al., 2017). Consequently, the RO method offers benchmark-quality temperature data and high-quality information on geopotential height fields (Steiner et al., 2013).

To investigate the width of the tropics, a variety of different methods were developed and tested, commonly referred to as

tropical width metrics (Adam et al., 2018; Davis and Birner, 2017). The primary purpose of these metrics is to determine the poleward edges of the Hadley cell and their changes in response to climate change. Many metrics exist, which can be more generally divided into direct and indirect measures of the circulation edges. The most common cell edge metric is the meridional mass stream function (PSI), which is directly connected to the Hadley cell (e.g., Grise and Davis, 2020; Staten et al., 2018). Other metrics focus more on tropical tropospheric features, which are connected with the Hadley circulation and

hence represent more indirect measures (Waugh et al., 2018). These include the subtropical jet (STJ), tropopause break (TPB) and eddy driven jet (EDJ). The correlations and interconnections between individual metrics have been of interest (Davis and Birner, 2017; Davis and Rosenlof, 2012), with a recent focus on their specific association with the Hadley cell. For instance, Menzel et al. (2019, 2024) found a disconnect between the PSI and the STJ and thus argue that the STJ should not be used as a Hadley cell metric. Furthermore, the TPB correlates with the STJ (Davis and Birner, 2017; Waugh et al., 2018), which

additionally questions the applicability of TPB associated metrics to determine the Hadley cell edges. As a counter example, although the EDJ lies outside the Hadley circulation and the tropical belt, changes of the EDJ metric correlate well with the ones from the Hadley cell edges (Menzel et al., 2023; Waugh et al., 2018). In accordance, the metric selection must be considered carefully in respect to the research question.

Initial studies exist analysing the tropical width using RO data (Ao and Hajj, 2013; Davis and Birner, 2013). Subsequent

research has shown that the high-quality temperature record from RO can be used to investigate zonally averaged as well as regional structures (Anjana et al., 2023; Luan et al., 2020; Mathew and Kumar, 2018). A main focus hereby has been on temperature-based metrics such as the TPB. Other variables and metrics from RO are still relatively unexplored. Most recently, there has been progress in deriving wind fields from RO geopotential height data (Danzer et al., 2024; Nimac et al., 2025b)



and further beyond the fundamental geostrophic and gradient wind (Nimac et al., 2025a). These new higher-order wind
estimates open the possibility to investigate wind-based metrics, such as the STJ, EDJ and possibly the PSI metric.

In this study, we test the potential of RO temperature and wind data to investigate the tropical width and its changes based on selected common tropical width metrics, including TPB, EDJ, STJ, and PSI. We validate RO in respect to most recent reanalysis datasets, inspecting the spatial and temporal consistency of a monthly timeseries. We separately test zonal-mean tropical width metrics and zonally resolved tropical width metrics, using high-quality temperature data and new higher-order
climatic wind fields, while also comparing different modern reanalyses in the process.

This paper unfolds as follows: A detailed description of the RO and reanalysis data, as well as an overview of the used metrics and statistical methods are given in Sect. 2. In Sect. 3, we start with a view on the zonally averaged results and continue with an analysis of the zonally resolved structure of the selected metrics. Subsequently, we provide an outlook on the necessary record length for trend estimates to outline the full potential of the future RO record. Finally, discussions and conclusions are
addressed in Sect. 4.

## 2 Data and methods

### 2.1 GNSS-RO data

In this study, monthly-mean RO multi-satellite temperature and wind climatology data is analysed. These are computed from RO phase data at UCAR/CDAAC (University Corporation for Atmospheric Research/COSMIC Data Analysis and Archive
Center), which provides data from six satellite missions: CHAMP (Wickert et al., 2001), GRACE (Beyerle et al., 2005), FORMOSAT-3/COSMIC (Anthes et al., 2008), C/NOFS (de La Beaujardière, 2004), SAC-C (Hajj et al., 2004), MetOp (Luntama et al., 2008). These satellites produce an average of 60 000 profiles per month (Angerer et al., 2017). The phase data is processed with the Wegener Center (WEGC) Occultation Processing System OPSv5.6, providing high-quality temperature and geopotential height data on altitude levels (Angerer et al., 2017; Steiner et al., 2020a), which are mapped to pressure levels.
Climatological wind fields are derived, based on Danzer et al. (2024) and Nimac et al. (2025b).

The RO data used in this study are derived with a moist air retrieval, combining ECMWF short-range forecast data with RO observations in the lower to mid troposphere (see Steiner et al. 2020, Table 1). The monthly-mean data are computed on a 2.5° x 2.5° latitude x longitude grid on 137 pressure levels, from 1000 hPa to 10 hPa. The horizontal grid is established using spatial and temporal weighting of the RO profiles. Hereby, the spatial weighting uses an area of influences which corresponds to a
constant 600 km radius, where the profiles are weighted according to their distance to the center location applying a bivariate (latitude-longitude) Gaussian function. The function has a peak at the center and a respective standard deviation of 150 km (300 km) in the latitudinal (longitudinal) direction. The temporal weighting uses Gaussian-time weighting within ±2 days (see further details in Ladstädter et al., 2022; Yessimbet et al., 2024). The multi-satellite record spans from 2001 to 2020. However, we chose to only study the period September 2006 to November 2020, due to fewer satellite missions in the early stages and
hence a lower number of occultation events, increasing sampling and statistical errors (Angerer et al., 2017; Scherllin-Pirscher



et al., 2017; Steiner et al., 2020a). Compared to initial RO studies of the tropical width (e.g., Ao and Hajj, 2013; Luan et al., 2020), we use a finer horizontal grid, which is possible due to the high number of RO profiles since 2006. This allows for a more detailed spatial investigation.

## 2.2 Reanalysis data

Three state-of-the-art reanalyses are selected, those are: ERA5 (Hersbach et al., 2020) by the European Center for Medium-Range Weather Forecasts (ECMWF), MERRA-2 Analysis (Gelaro et al., 2017) by the National Aeronautics and Space Administration (NASA) and JRA-3Q (Kosaka et al., 2024) by the Japanese Meteorological Agency (JMA). Just like with the RO dataset, we use temperature and zonal wind reanalysis data on isobaric levels to calculate various tropical width metrics, as well as the meridional wind to analyse the atmospheric mass stream function. The horizontal resolution, vertical range,

number of pressure levels and the time period studied for each dataset are listed in Table 1. For comparison with the RO record, the monthly reanalysis data are downloaded on their native grid and then regridded using Climate data operators (CDO) bilinear interpolation function "remapbil" (Schulzweida, 2023) to match the 2.5° x 2.5° latitude x longitude grid of the RO data. We note that the individual reanalyses refer to the interpolated version, unless stated otherwise.

**Table 1: List of datasets and their associated information used in this study.**

| Data Product | Institution | Horizontal resolution (lat×lon) | No. of pressure levels | Vertical range (hPa) | Time range (studied) |
|---|---|---|---|---|---|
| RO OPSv5.6 | WEGC | 2.5°×2.5° | 137 | 1000–10 | 2006–2020 |
| ERA5 | ECMWF | 0.25°×0.25° | 37 | 1000–1 | 1980–2024 |
| MERRA-2 ASM | NASA | 0.5°×0. 625° | 42 | 1000–10 | 1980–2024 |
| JRA-3Q | JMA | 0.375°×0.375° | 45 | 1000–0.01 | 1980–2024 |

## 110  2.3 Tropical edge metrics

To quantify the tropical width, we analyzed five different metrics using zonal-mean and zonally resolved tropical width metrics:

1)  TPB (max $\partial Z/\partial \varphi$) is the latitude where the meridional tropopause height gradient has its maximum between 0° and 60° latitude. The tropopause height is defined according to the World Meteorological Organization (WMO, 1957) as "the

lowest level at which the lapse-rate decreases to 2°C/km or less, provided that the average lapse-rate between this level and all higher levels within 2 km does not exceed 2°C/km". The tropopause features a height drop between the tropics and extra-tropics, which has previously been applied to define the tropical edge (e.g., Davis and Rosenlof, 2012).

2)  TPB (max $\Delta\theta$), also known as the maximum dry bulk static stability, estimates the latitude of the maximum potential temperature difference at the tropopause and the minimum value of the air column (Davis and Birner, 2013, 2017). Again,

this metric is defined between the equatorial boundary 10° and polar boundary 60°. Although the RO dataset based on moist-air retrieval is available from 1000 hPa upward, the data is only observation-dominated at higher levels, while



information from ECMWF forecasts comes in at lower levels (Steiner et al., 2020a). Therefore, we specifically select the minimum value for RO at the 400 hPa level following Luan et al. (2020). For reanalysis the minimum value is located near the surface. Previous comparisons have shown good agreement with some seasonal exceptions (Luan et al., 2020).

3)  EDJ is located at the latitude of the zonal wind maximum closest to the 850 hPa level poleward 15° and equatorward 70° (Adam et al., 2018). For RO data, the metric is adapted to select the maximum at the 700 hPa level. As described for TPB (max $\Delta\theta$) metric above, RO provides independent high-quality information at higher levels of the troposphere, where the data is purely observation-based and not combined with ECMWF forecasts. This specific level is at the upper range previously used to study the EDJ (Keel et al., 2024; Woollings et al., 2010), hence additional tests are provided. Figure 1

shows generally good agreement for the tested levels for zonal-mean and zonally resolved results, and thus supports the decision to use the 700 hPa level for RO. Over Europe and Eastern Asia, the metric deviates for the individual levels and hence should be interpreted with care.

4)  STJ describes the latitude of the subtropical jet core, which is identified at the zonal wind maximum between 100 hPa to 400 hPa after subtracting the zonal wind of the EDJ. The STJ is defined in the latitude range from 10° to 60°.

5)  PSI is defined as the zero crossing of the meridional mass stream function at 500 hPa between 30° to 60° latitude. The zonal-mean PSI metric is calculated according Eq. (1):

$$\psi = \frac{2\pi a cos\varphi}{g} \int_0^p \bar{v} dp \,, \tag{1}$$

where $a$ is Earth's radius, $\varphi$ is the latitude, $g$ is the gravitational acceleration, p is the pressure and $\bar{v}$ denotes the zonal-mean meridional wind. For the longitudinally dependent metric, first the meridional component of the divergent wind $v_{div}$

is calculated using the windspharm software package (Dawson, 2016), then the longitudinally dependent stream function $\psi$ (Galanti et al., 2022) is calculated using to Eq. (2):

$$\psi = \frac{1}{g} \int_0^p v_{div} dp. \tag{2}$$

The Tropical-Width Diagnostic software package for Python (PyTropD) builds the foundation for this metric calculations (Adam et al., 2018). Although this package was originally developed for computing zonal-mean metrics, by now it has also

been tested for zonally resolved results. Luan et al. (2020) investigated the regional tropopause, whereas Liu et al. (2021) analysed the jet streams at individual longitudes. As described above, the computation of the zonal-mean mass stream function and the zonally resolved mass streamfunction is not identical. For the regional streamfunction, we build on initial work of Schwendike et al. (2014) and later studies by e.g., Galanti et al. (2022).





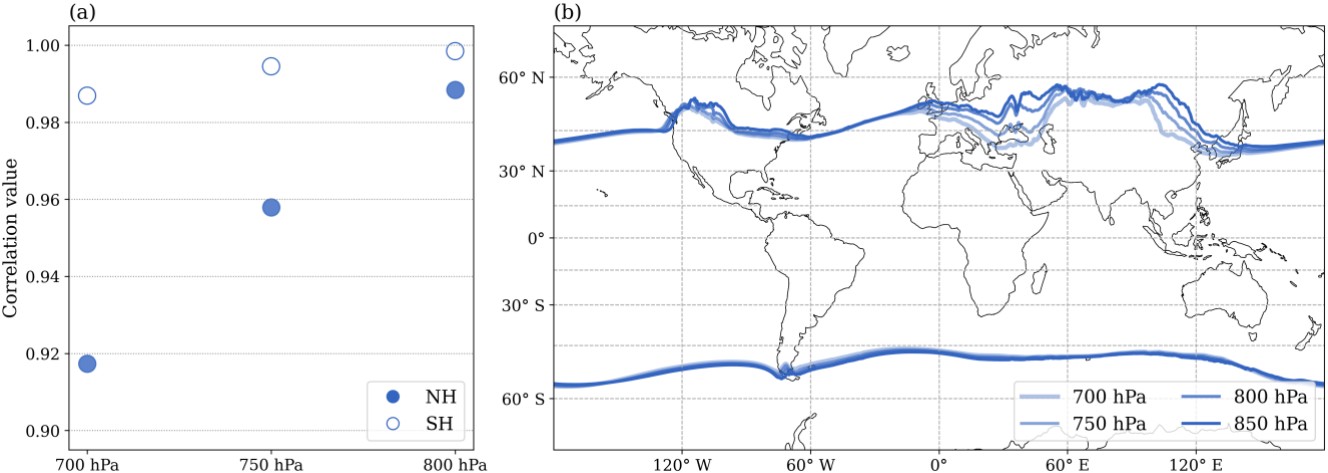

Figure 1: Results of the EDJ metric at different pressure levels for monthly ERA5 values averaged over the period 1980 to 2024 on its native grid, (a) shows correlation values for 700 hPa, 750 hPa and 800 hPa with the commonly used 850 hPa level, (b) depicts the zonally resolved metric.

**2.4 Additional methods**

To compare the metric results from RO and reanalyses, statistical methods are applied. A correlation is performed on monthly-mean anomaly values. The metric results over time are linearly interpolated to fill missing values, this is e.g., the case for the PSI metric over the summer months when the Hadley cell is highly asymmetric. Statistical significance of a correlation is assumed for p-value < 0.001 (i.e., 99.9% confidence interval).

**3 Results**

In this section, we test the suitability of RO data to (i) calculate zonal-mean tropical width metrics and (ii) investigate the regional structure of the selected metrics. Hereby, RO will be validated using reanalyses. Finally, we aim to (iii) provide a perspective on the necessary record length for trend studies using these metrics, to emphasize the future value of the long-term consistent RO record.

**3.1 Zonal-mean metrics**

In an initial analysis we study the monthly time series of the selected zonal-mean temperature and wind metrics over the RO record period September 2006 to December 2020 (Fig. 2). The time series compares RO data and multiple reanalysis, providing a first insight for the northern and southern hemisphere, hereafter NH and SH, respectively. The values from RO show good agreement with values from reanalyses for the TPB metrics and STJ metric, following the same patterns over time. Contrarily, while the EDJ appears to follow a similar seasonal cycle for all datasets, the absolute position of this metric differs for RO. The results of the EDJ from RO, in comparison to the reanalysis datasets, are shifted towards the equator particularly on the





NH. This could be a result of the chosen method, as explained in Sect. 2.3, the EDJ for RO is calculated at 700 hPa, in contrast to the reanalyses where this metric is calculated at 850 hPa. An extended discussion on the time series analysis of these metrics using RO data is done in Davis and Birner (2013). They investigated the seasonal cycle and interannual variability of the Hadley cell. However, their study is based on a rather short RO record from 2007 to 2011 using data from the COSMIC mission.

**Figure 2:** Time series of five different metrics (a-e) describing the monthly latitudinal edges of the tropical width (TPB, STJ) and Hadley cell (PSI), as well as the location of the eddy driven jet (EDJ) from September 2006 to December 2020. The NH and SH edge is shown for the RO and reanalysis datasets. The range of the vertical axis is chosen for the respective results.





To better quantify the validity of the RO dataset, correlations between RO and reanalyses, between different reanalyses, and
between reanalyses on different grid spacings are presented in Fig. 3. Figure 3a shows the correlation results between RO and
the three state-of-the-art reanalyses. Hereby, the TPB metrics, STJ metric and EDJ metric have strong correlations. Especially
the jet metrics on the SH are in very good agreement between the datasets, which indicates the high quality of the derived
zonal wind component from RO. The correlations with reanalyses are clustered very close together for the individual metrics
and hemispheres, with the exception of TPB (max $\partial Z/\partial\varphi$). For this metric, the correlation values between RO and MERRA-
2 are noticeably lower which differs from the results of the other reanalyses. Overall, the high corelation values provide a first
impression that RO is suitable for determining these zonal-mean tropical width metrics.

A first attempt to calculate the mass stream function using RO winds accounting for advection terms was performed. However,
the results show that the underlying wind field estimates from RO geopotential height are at this point not qualified to determine
the Hadley cell edge using the PSI metric. In the NH (in Fig. 2e), the calculated results appear to be in range of the Hadley cell
edge as determined by reanalyses, while in the SH, the wind estimates of the meridional wind component seem to lack the
necessary detail to calculate the PSI metric. Figure 3a supports these findings, as no correlation can be found between RO and
reanalyses on the SH. On the NH, the correlation is very weak. However, we note that this does not speak against the overall
quality of the higher order winds, rather the zonal-mean meridional wind is a small term, which can so far not be captured
accurately with the applied wind equations. For further details on the underlying wind data see Nimac et al. (2025b). Overall,
our results demonstrate a great improvement of the winds calculated from RO data (Scherllin-Pirscher et al., 2014), as the
zonal average of previous wind estimates, such as geostrophic and gradient wind, was by default zero and thus the zonal-mean
stream function could not be calculated at all. In this regard, we highlight the importance of advancements in the development
of global, observational winds to validate reanalysis and climate model data, which has also previously been emphasized by
others (Pikovnik and Zaplotnik, 2025). As a conclusion from the analysis in Fig. 2, the PSI metric calculated from RO winds
will not be further investigated in this study. Results from reanalysis data are still included in our investigation, as the Hadley
cell edge represents a very important feature of the tropical atmosphere, and the PSI is one of its most common metrics.

A critical factor that must be taken into account is the role of grid dependence. The correlation presented in Fig. 3a is performed
for RO and reanalyses interpolated on the same grid (i.e., 2.5° x 2.5°), which is rather coarse compared to the native grid of
the reanalysis fields. As a result, it could be argued that there is a loss of information. The grid dependence of the specific
metrics and in respect to the studied dataset has been previously discussed. Davis and Birner (2016) showed that the horizontal
resolution is of relevance in trend studies of the tropical width, as a finer grid can result in a narrower tropical belt. Furthermore,
the spread in trend results from various climate models can be attributed to their different grid sizes. Another study by Luan et
al. (2020) based on reanalysis, showed that for various TPB metrics on a regional perspective, the structure of the tropopause
remained the same for multiple tested grids. This information will be further important in Sect. 3.2.
Here we provide further information on the relevance of the grid spacing of the underlying dataset for these specific metrics.
Overall, the methods applied in this study use either linear interpolation between two data points or a function to determine
the weighted center, to reduce the grid dependence of the metrics (Adam et al., 2018). Figure 3b illustrates the grid dependence




of each metric for reanalyses. Generally, the correlation between the native grid and interpolated grid for all reanalyses show good agreement, which indicates that the results from the datasets on their native grid and lower resolution grid are nearly

identical. This strengthens the confidence in the RO datasets and underlines that the results provided can be highly relevant despite the lower grid resolution. Nevertheless, two things stand out from this plot. One aspect is, TPB (max $\partial Z/\partial\varphi$) on both hemispheres has lower correlation values than the other metrics, seen for all datasets. Another aspect regards the PSI metric on the NH. For this metric, the correlation is smaller for ERA5 and MERRA-2, whereas the correlation values stay high for JRA-3Q. This is the only instances where we see a difference between the individual datasets. In any way, the results from

both TPB (max $\partial Z/\partial\varphi$) and PSI on the NH depict a strong correlation, although these metrics appear to be more sensitive to the underlying resolution of the dataset.

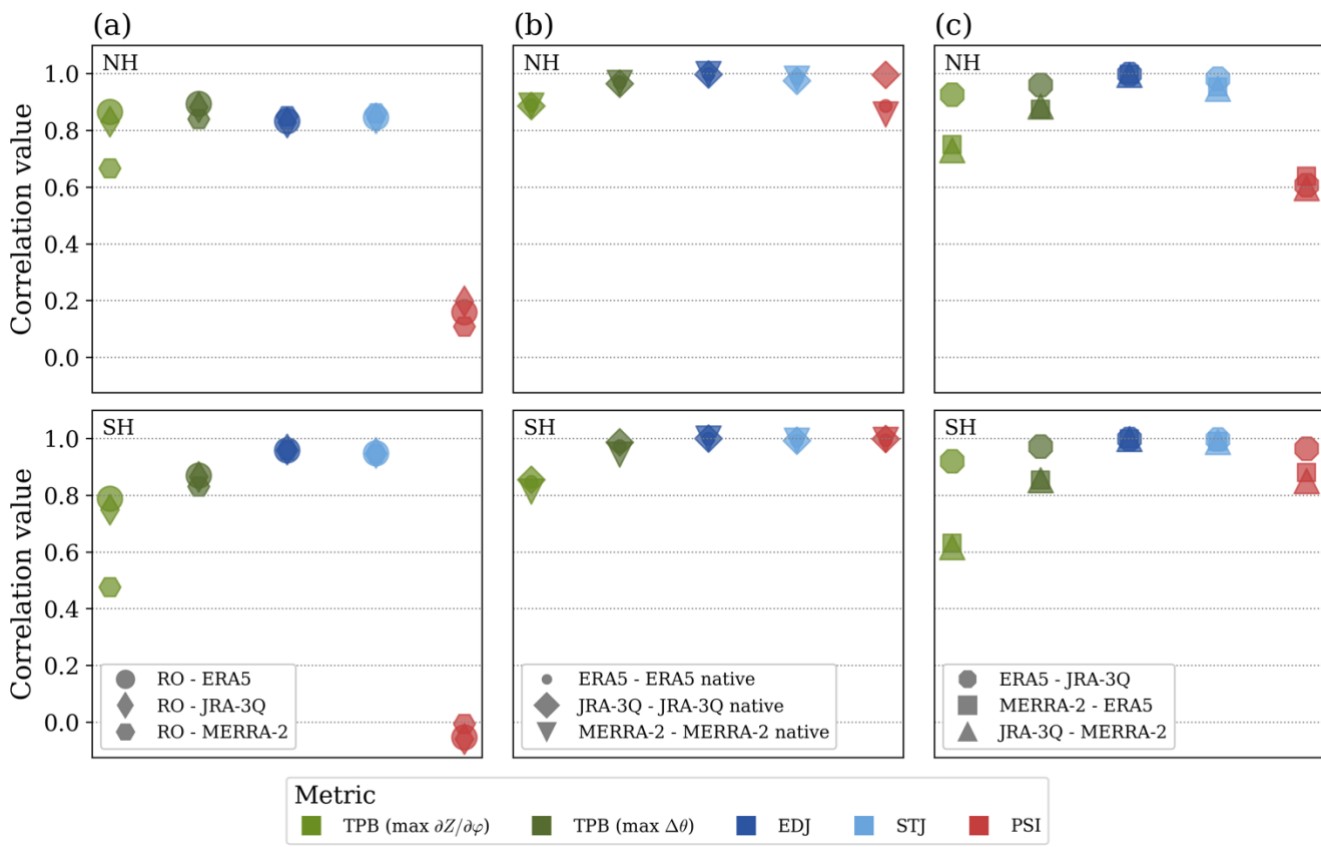

**Figure 3: Correlation of monthly anomalies of tropical edge values between the different datasets for five different metrics over the time period September 2006 to December 2020. The correlation of RO and different reanalyses as shown in (a), (b) reveals the**
**correlation of the reanalyses interpolated to the RO grid and on their native grid, and (c) depicts the correlation among the different reanalysis datasets. All correlations are statistically significant (p-value < 0.001), apart from the PSI metric in (a).**

Figure 3c visualizes the correlation values of the five metrics between the different reanalysis datasets. The results show overall strong correlations with no specific differences between the hemispheres for the TPB and jet metrics. Contrary, the PSI metric displays a greater correlation on the SH in comparison to the NH. Consistent with the correlations between RO and reanalyses



(i.e., Fig. 3a), for TPB (max $\partial Z/\partial\varphi$) the correlations among the individual reanalyses are slightly lower than for the other metrics. The correlation for this metric improves for the reanalyses on their native grid (Fig. A1). Generally, for both TPB metrics, ERA5 and JRA-3Q, which are newer datasets, are considerably stronger correlated among each other than towards MERRA-2. It is also noteworthy that the correlation results of the jet metrics are approximately equal to one, which suggests that the individual reanalysis are very similar.

**3.2 Regional structure of the metrics**

Next, we evaluate the utility of RO data to examine the regional structure of the TPB and jet metrics. Figure 4 presents the results of the longitudinally resolved metrics for the NH and SH averaged over the inspected period, September 2006 to December 2020, for RO and reanalyses. The plot shows differences in the regional structures and the location of these metrics. An additional plot (Fig. A2) provides information for reanalyses on their native grid, showing that the overall structure of the
metrics remains.

The TPB metrics (Fig 4a-b) display a mostly uniform structure around the globe, where TPB (max $\partial Z/\partial\varphi$) is located slightly more poleward than TPB (max $\Delta\theta$), which position varies around 30° on each hemisphere. In contrast, the jet stream metrics reveal a stronger regional variance. Figure 4c and 4d show two distinct jets over all longitudes, although we note that for shorter time steps the jets can merge into one at particular longitudes and for some seasons (Lee and Kim, 2003). The STJ
(Fig 4c) on the NH is located around 15 °N in the eastern Pacific and over the Atlantic, and at about 30 °N over America, Asia and the western Pacific. On the SH the STJ is rather uniform at 30 °S, with an exception over the eastern Atlantic. The EDJ (Fig 4d) is found more poleward compared to the STJ. On the NH the EDJ varies around 45 °N and on the SH it is located even further towards the pole at about 50 °S. The structure of the EDJ seems to be influenced by the topography, explicitly on the NH. Finally, Fig 4e, shows the PSI metric for reanalyses. The mass stream function forms sub cell structures, which can
vary in their extend and strength and show a clear seasonality (Li et al., 2022). As a result, this metric is not always globally available. The PSI displays a relatively uniform edge on the NH, which stretches from the mid Pacific to the Tibetan Plateau. On the SH, there is more variation. In general, we note that our analysis focuses on monthly values averaged over a longer time period. Thus, we are not explicitly regarding the seasonal cycle of the metrics, although the location and structure of the seasonal metrics can differ (see e.g., Li et al., 2022; Luan et al., 2020).
Below each map showing the regional structure of each metric, a difference plot more clearly visualizes the specific difference between RO and the reanalyses datasets. Overall, the metric results from different datasets align well. Some exceptions occur over the NH. In general, the temperature-based metrics show closer agreement then the wind-based jet metrics and the differences for all metrics are smaller over the SH than on the NH. The hemispheric differences could be a result of the underlying topography, as on the SH, there is generally less land surface and mountain ranges which can influence the position
of the specific tropospheric features.




**Figure 4: Mean longitudinally varying tropical edge of the NH and SH for RO and reanalyses for five cell edge metrics (a-e) over the RO period. Additionally, the difference between RO and reanalyses is shown for each edge, with the land surface indicated in grey shading. The PSI-metric (e) is only shown for reanalyses.**





Furthermore, it is hypothesised that the differences might be smaller over the oceans compared to over land surfaces, due to the available number of observations assimilated in reanalyses. More precisely, reanalyses assimilate observational data from various sources. However, in assimilating many observational data sources, reanalyses introduce spatial and temporal inhomogeneities. One important observational data source for reanalyses are radiosondes, which are mostly available over the land surface (Fujiwara et al., 2017). This can lead to these theorized differences and could also contribute to possible land-sea

contrasts. The hypothesis cannot be uniformly confirmed among the different metrics. While TPB (max $\Delta\theta$) indeed shows larger differences among RO and reanalysis over land, TPB (max $\partial Z/\partial\varphi$) has the greatest deviations above the ocean. As for the jets, the deviations between RO and reanalyses appear to be connected to the topography. In regions with high mountain ranges, such as the Rocky Mountains and Tibetan plateau, it is difficult to determine these metrics.

**Trend perspectives**

Finally, we provide a perspective on the needed time length to calculate trends, in order to emphasize the possible future value of the long-term consistent RO record for trend studies. In this regard, we investigate the three reanalysis datasets over their longer record (i.e., 1980 to 2024). Figure 5 shows all possible trend values for a given time window length (i.e., 15-, 25-, 35 years) over the reanalysis record period for the respective datasets, metrics and hemispheres. Additionally, the median of all calculated trends for a specific time window and dataset is marked, to indicate the overall trend of the metric.

Figure 5 demonstrates the importance of a sufficient record length for trend analysis of tropical widening. Previously, also shorter data records, such as the RO record, have been used in trend studies of the tropical width, providing varying results (Ao and Hajj, 2013; Darrag et al., 2022). Figure 5 clearly visualizes the wide spread of trend for these shorter records. For instance, 15-year trends over the 1980 to 2024 time period range from poleward to equatorward shifts for the metrics on both hemispheres. Particularly, TPB (max $\Delta\theta$) and EDJ show a larger spread of the calculated trend signals. The inconclusive trend

results continue also for the 25-year periods, with the exception of TPB (max $\Delta\theta$) and PSI on the SH. Besides these, the trend value distribution only starts to support the general understanding of tropical widening, for the longest time window length in this study, i.e., 35 years. Here, the trends for most datasets and metrics are unilaterally placed relative to zero, indicating a significant poleward shift of the metric. This is denoted by positive trend values on the NH and negative trend values on the SH.

Generally, there are differences between temperature and wind-based metrics. While the wind-based metrics exhibit a greater spread of trends on the NH, temperature-based metrics show more spread of trends on the SH. Furthermore, the distribution and range of the trends for the temperature-based metrics differ more between the datasets. Interestingly, the STJ and EDJ metric, which are both based on zonal wind, showcase very similar results between the different datasets. The spread of the single trends for a time window length and their distribution is consistent between all datasets. This is unlike the other metrics,

which are based on temperature data (i.e., TPB) and the meridional wind (i.e., PSI), where the results of the individual datasets visibly differ. Furthermore, the STJ and EDJ trend results on the SH are more consistent with less spread than on the NH, where metrics over high mountain regions may add to the greater spread.





We note that the broad distribution of the trend values for a time window length might also indicate the strong influence of climate variability and other drivers of tropical width change, obscuring the global warming signal. Multiple drivers, such as

internal climate variability and natural and anthropogenic external forcings, influence the location of these tropical width metrics, as discussed in recent literature (Grise et al., 2019; Staten et al., 2019). These effects also complicate the estimation of tropical widening trends and underline the importance of a sufficiently long data record.

**Figure 5: Cell edge trend for different metrics (a-e) and reanalyses (shading) for the three time window lengths 15-, 25-, 35-years.**

**Every trend of a specific time length calculated within the period 1980–2024 is indicated by a horizontal line. A shaded bar is in the background of each dataset for the different time windows. The median of each dataset and specific time window length is indicated by a longer horizontal line.**



Besides the record length also the long-term consistency of the used data set is significant for trend assessments, which is a known flaw of reanalysis (Fujiwara et al., 2017). In this regard, one surprising and interesting result can be seen in Fig. 5a.

The results of MERRA-2 on the SH indicate an equatorward shift, which is contrary to the other datasets and metric results. No information on a similar discovery could be found in the literature, as to the best of our knowledge, a trend for this particular metric over the available record length (i.e., 1980 onwards) has not been investigated for MERRA-2. Additional testing indicates a possibly, artificially induced trend which results from an unexpected change of the TPB (max $\partial Z / \partial \varphi$) time series. More specifically, when looking at the longer time series of the TPB (max $\partial Z / \partial \varphi$) in comparison between the different

reanalysis datasets, Fig. 6 suggests that the shift might have occurred in the early 21$^{st}$ century. This potentially implies a possible discontinuity of the MERRA-2 temperature record near the tropopause height. In this time period multiple changes in the data assimilation of the reanalysis have occurred, e.g., GNSS-RO bending angles since July 2004, MLS temperature from August 2004 and ATOVS starting in May 2007, are being assimilated in MERRA-2 (Fujiwara et al., 2017; Gelaro et al., 2017). Based on these current findings, a long-term trend of the TPB (max $\partial Z / \partial \varphi$) metric in MERRA-2 should be viewed

with caution and a more detailed study on these results would be of interest.

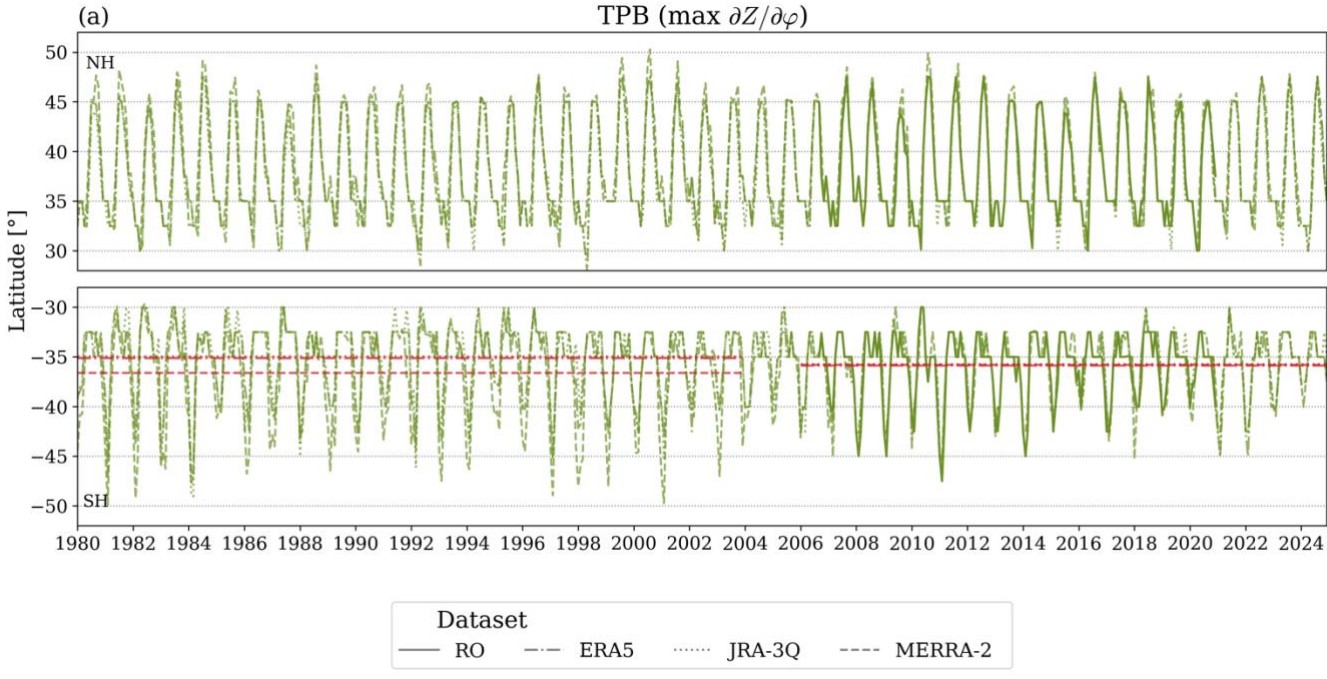

**Figure 6: Monthly time series of TPB (max $\partial Z/\partial \varphi$) latitudinal edges from 1980 to 2024 (a). The NH and SH edge is shown for the RO and reanalysis datasets. The red horizontal lines on the SH indicate the mean edge value for the three reanalyses spanning over two different time periods.**





## 4 Conclusions and discussion

In this study we demonstrated the applicability of observational RO temperature and new RO wind data to examine a range of tropical width metrics, including tropopause break and jet stream metrics. The study focused on the potential of RO data for zonal-mean and zonally resolved positions of selected tropospheric features using common tropical width metrics. By showing similarly strong correlations between RO and reanalyses, as demonstrated between different reanalyses, our results illustrate the capability of this a globally available, independent observational dataset.

Our study results based on the WEGC RO OPSv5.6 record and most recent reanalyses are consistent with previous findings of RO records, the studied TPB metrics show high agreement between the datasets (e.g., Darrag et al., 2022; Luan et al., 2020). Furthermore, we extend upon earlier temperature-based studies, additionally focusing on wind-based metrics and using new wind records from RO. The recent development of these advanced RO wind fields beyond the fundamental geostrophic approach (Nimac et al., 2025a), allows a more detailed investigation of the jet streams. Additionally, this study is the first of its kind, analysing also regional jet streams based on RO data. The results of the EDJ and STJ from RO agree with the results from the three reanalyses. Especially on the SH the results of the jet streams show very close agreement between the datasets, highlighting RO as an important observational data source, where others, e.g., radiosondes are scarce.

We have proven the general applicability of RO, however, further methods to define the jet streams exist (Keel et al., 2024; Liu et al., 2021), which were not tested in the process. Hence, a detailed investigation of the jets using RO observations presents an interesting direction for future research.

Moreover, a focus of this work has been on the grid dependence of the specific metrics. The RO record studied here is available on a 2.5° x 2.5° grid. Despite this being a finer spatial grid than used in previous RO-based studies, it is rather coarse in comparison to state-of-the-art reanalyses. This issue was tested by comparing results from native and regridded reanalyses. The findings are fairly consistent also for the reduced grid spacing in the zonal-mean and for zonally resolved data, as shown also in Luan et al. (2020) for TPB metrics.

While we see great potential in RO records for the study of tropical atmospheric features, we also acknowledge their constraints. RO data has been successfully applied in climate studies, for instance, to investigate temperature trends (e.g., Ladstädter et al., 2023; Steiner et al., 2020b) and tropopause trends (Meng et al., 2021) over recent decades. However, for trend studies of the tropical width, which exhibits large variability, the record length is not yet considered long enough using regionally resolved RO data since 2006. This currently confines the full potential of the RO dataset. Consistent with other studies (e.g., Grise et al., 2018), we showed that an extended record length is necessary for trend estimation of these tropical width metrics.

To summarize, the obtained results show the potential of RO data for the study of tropical width metrics, highlighting a clear set of applicable methods for this specific observational dataset. This paves the way for future trend studies as the observational record grows. Our investigation extended beyond the well-established RO temperature record to include advanced RO wind information. While we perceived strong potential in RO data for temperature-based metrics and zonal-wind-based metrics, we





find that the computation of the meridional RO wind component needs further research. Nevertheless, results confirm a high suitability for zonal-mean applications and zonally resolved studies. The RO record is highly relevant in validating reanalysis

and climate model data, when the full potential of its long-term consistency and high accuracy can be employed.

**Appendix**



**Figure A1: Correlation of monthly anomalies of tropical edge values between the different datasets for five different metrics over the time period September 2006 to December 2020. The correlation of RO and different reanalyses on their native grid as shown in**
**(a), and (b) the correlation among the different reanalysis datasets on their native grids. All correlations are statistically significant (p-value < 0.001), apart from the PSI metric in (a).**



**Figure A2: Mean longitudinally varying tropical edge of the NH and SH for RO and reanalyses for five cell edge metrics (a-e) over the RO period. The PSI-metric (e) is only shown for reanalyses. Bold lines show the results from the datasets on the 2.5° x 2.5° latitude x longitude grid, while shaded lines show the results from the reanalyses on their native grid.**

## Data and code availability

The WEGC RO OPSv5.6 data used in this study are archived under https://doi.org/10.25364/WEGC/OPS5.6:2021.1 (EOPAC Team, 2021). All reanalyses data are publicly available: ERA5 was downloaded from the climate data store (https://climate.copernicus.eu, last access: 29 July 2025, https://doi.org/10.24381/cds.6860a573), MERRA-2 was acquired from GES DISC (https://disc.gsfc.nasa.gov, last access 29 July 2025, https://doi.org/10.5067/V92O8XZ30XBI), JRA-3Q was obtained from NSF NCAR Research Data Archive (https://rda.ucar.edu, last access 29 July 2025, https://doi.org/10.5065/PH0D-MH18). The calculations were performed using Python 3.11 and build on the freely available software packages windspharm (Dawson, 2016) and PyTropD (Adam et al., 2018).



**Author contribution**

Conceptualization: AR, JD; Data curation: AR; Formal analysis: AR; Funding acquisition: JD; Methodology: AR, JD; Supervision: JD, AKS; Validation and visualization: AR, JD, AKS; Writing (original draft preparation): AR, JD; Writing (review and editing): AR, JD, AKS.

**Competing interests**

The authors declare that they have no conflict of interest.

**Acknowledgements**

We thank the UCAR/CDAAC RO team for providing the RO excess phase and orbit data, and the WEGC RO team for supplying the OPSv5.6 retrieved profile data. We are particularly grateful to Florian Ladstädter (WEGC) for providing the monthly gridded climatology data. Within the Strato-Clim team, we thank Johannes Unegg (WEGC) and Irena Nimac (WEGC) for providing the wind fields, and we especially thank Gottfried Kirchengast (WEGC) for many fruitful discussions. Finally,
we gratefully acknowledge the Austrian Science Fund (FWF) for supporting this work under grant no. P-40182, and the University of Graz for the financial support of the publication costs (open access).

**Financial support**

This research has been supported by the Austrian Science Fund (project Strato-Clim, grant no. P-40182).

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
