# Peer review of "The potential of GNSS radio occultation data for the analysis of the tropical width: a comparison with reanalyses"

_EGUsphere, 2025_

## Author Comment (AC1)

**Response to Anonymous Referee #1**

Referee comments: https://doi.org/10.5194/egusphere-2025-3745-RC1

Manuscript: Reiter, A., Danzer, J., and Steiner, A. K.: The potential of GNSS radio occultation data for the analysis of the tropical width: a comparison with reanalyses, EGUsphere [preprint], https://doi.org/10.5194/egusphere-2025-3745, 2025.

The authors leverage 15 years of GNSS radio occultation (RO) temperature profile data to examine the width of the tropics and its change over time. They compare the resulting diagnoses to longer records from several state-of-the-art reanalyses. Ultimately, it is demonstrated that RO data provide useful characterizations that broadly agree with the reanalysis diagnoses, especially upper troposphere lower stratosphere metrics where RO data are complete and most reliable. While I find the study to be mostly well-constructed and detailed, there are a few aspects that require a bit more clarification which I outline below.

We would like to thank the reviewer for the constructive assessment, for generally finding our study interesting and well-constructed, and for the helpful comments for further improvement. We carefully considered and answered all comments below (comments are quoted in *italic with gray background*, with the responding answers below each comment). Line numbers refer to the original manuscript.

**General Comments**

**1 There are many instances of "on the NH" or "on the SH" that should all be revised to "in the NH" or "in the SH".**

Thank you for noticing it. We corrected it throughout the manuscript.

**Specific Comments**

**2 Line 7: the opening sentence of the abstract is a too strong of a statement. As the authors acknowledge later, this result is contingent upon the metric used. The language should be softened here.**

Thank you for pointing this out. Indeed, the opening sentences was a bit strong, we refined it to:

L7: "The tropical width is changing, with a poleward expansion being linked to anthropogenic climate change."

**3 Line 29: delete unnecessary period after "include"**

Thank you for noticing it. We corrected it in the text.

**4 Line 32: "systems" should be "system"**

Thank you for noticing it. We corrected it in the text.

**#5 Lines 71-75: these sentences are entirely unnecessary**

While we agree, that the paragraph includes too many details, we still want to keep a brief overview of the structure of the manuscript. For that reason, we rephrased and shortened the paragraph in the following way:

L71: "This paper is structured as follows: The datasets and methodology are detailed in Sect. 2, while Sect. 3 presents the results for the selected metrics based on RO and reanalysis data. Finally, discussions and conclusions are addressed in Sect. 4."

**6 Line 85: a brief discussion of the wind retrieval is warranted here. It is later stated that the wind isn't a simple geostrophic retrieval, so what is it?**

Thank you for this suggestion, we added the following sentences to chapter 2.1. "GNSS RO data" to provide more details on the wind retrieval. The method and application to RO data are explained in detail in Nimac et al. (2025) and Unegg et al. (2025). However, both manuscripts are currently under review and hence cannot be cited in the final version of this manuscript due to the submission guidelines:

L93: "As a novelty the winds are computed using a best-estimate algorithm which dynamically applies the most suitable wind retrieval method dependent on latitude and altitude. Thereby, the method uses the initial corresponding balanced wind estimates (i.e., the geostrophic equation in the troposphere and the gradient wind in the stratosphere) and adds advective contributions on top of these initial wind estimates. Furthermore, in the equatorial region curvature terms are included to the equatorial balanced winds."

**7 Lines 92-93: is a ±2 day Gaussian time-weighting approach appropriate? This could be better justified/explained.**

We added a sentence to better explain the approach.

L93: "Temporal and spatial weighting ensures that the observed information is fully utilized, while keeping the number of empty grid points low, with remaining gaps filled using bilinear interpolation (see further details in Ladstädter et al., 2022; Yessimbet et al., 2024)."

**8 Line 250: "extend" should be "extent"**

Thank you for noticing it. We corrected it in the text.

**9 Line 257: "then" should be "than"**

Thank you for noticing it. We corrected it in the text.

**10 Line 311-312: there are a few studies that have revealed this narrowing via tropopause break metrics — Martin et al. 2020, https://doi.org/10.1175/JCLI-D-19-0629.1; Zou et al. 2023, https://doi.org/10.3389/feart.2023.1177502; Turhal et al. 2024, https://doi.org/10.5194/acp-24-13653-2024**

We thank you for your insight and the references. While for example Martin et al. (2020) did look at trends of the tropopause break and also describe a narrowing trend for some longitudes, they specifically mention that they did not find a discontinuity in the dataset. However, our study builds on different presets, Martin et al. (2020) did not check the same specific TPB metric as we did. Nevertheless, we are thankful for the comment and added a sentence to improve clarity.

L311: "While in general, narrowing trends in various TPB metrics for MERRA-2 have been found in other studies (Martin et al., 2020; Zou et al., 2023), this specific difference in the results of the TPB ( $\max \partial Z/\partial \varphi$ ) has not been documented."

**11 Line 330: "this a globally" should be "this globally"**

Thank you for noticing it. We corrected it in the text.

**12 Line 335: delete unnecessary comma after "(Nimac et al., 2025a)"**

Thank you for noticing it. We corrected it in the text.

**References**

- Ladstädter, F., Stocker, M., Yessimbet, K., and Steiner, A. K.: GNSS RO providing a detailed view on the thermal structure and changes in Earth's atmosphere, International Workshop on Occultations for Probing Atmosphere and Climate 2022, Leibnitz, Austria 8–14 September 2022, https://static.uni-graz.at/fileadmin/\_files/\_event\_sites/\_opacirowg2022/programme/08.9.22/AM/Session 1/OPAC-IROWG-2022 Ladstaedter.pdf, 2022.
- Martin, E. R., Homeyer, C. R., McKinzie, R. A., McCarthy, K. M., and Xian, T.: Regionally Varying Assessments of Upper-Level Tropical Width in Reanalyses and CMIP5 Models Using a Tropopause Break Metric, Journal of Climate, 33, 5885–5903, https://doi.org/10.1175/JCLI-D-19-0629.1, 2020.
- Nimac, I., Unegg, J., and Danzer, J.: Climatic higher-order balanced winds beyond geostrophic and gradient wind fields, Earth and Space Science [submitted], 2025.
- Unegg, J., Nimac, I., and Danzer, J.: Beyond geostrophic and gradient wind: Enhancing the estimation of climatic wind fields from radio occultation, Earth and Space Science [submitted], 2025.
- Yessimbet, K., Steiner, A. K., Ladstädter, F., and Ossó, A.: Observational perspective on sudden stratospheric warmings and blocking from Eliassen–Palm fluxes, Atmospheric Chemistry and Physics, 24, 10893–10919, https://doi.org/10.5194/acp-24-10893-2024, 2024.
- Zou, L., Hoffmann, L., Müller, R., and Spang, R.: Variability and trends of the tropical tropopause derived from a 1980–2021 multi-reanalysis assessment, Front. Earth Sci., 11, https://doi.org/10.3389/feart.2023.1177502, 2023.

---

## Author Comment (AC2)

**Response to Anonymous Referee #2**

Referee comments: https://doi.org/10.5194/egusphere-2025-3745-RC2

Manuscript: Reiter, A., Danzer, J., and Steiner, A. K.: The potential of GNSS radio occultation data for the analysis of the tropical width: a comparison with reanalyses, EGUsphere [preprint], https://doi.org/10.5194/egusphere-2025-3745, 2025.

This paper describes the use of GNSS-RO data to study several indirect metrics of the tropical width. This includes the use of tropopause temperature (which has been done using RO before) and locations of the subtropical and eddy driven jets based on the newly derived winds fields obtained from the RO geopotential height data. The authors show that the RO results generally compared well with the reanalyses, although some systematic (especially zonally varying) differences were noted. The authors argued that the RO data period (2006 – 2020) was currently too small to detect the tropical width trends, with 35 years needed based on the reanalyses.

This is a well-written paper with interesting results on an important topic. I recommend its publication after the following comments are addressed.

Thank you very much for your positive and encouraging assessment, for generally finding our study well-written and your recommendation to be published after minor revision. We carefully considered and answered all comments below (comments are quoted in *italic with gray background*, with the responding answers below each comment). Line numbers refer to the original manuscript.

**Comments:**

**1: In Sec 2.1, the authors should provide some description on the accuracy of the RO data used in the analysis, especially the wind retrieval which is a fairly new product that's not known by most readers**

We are grateful for this comment, adding information on the accuracy of the RO dataset is very important to highlight the underlying data quality of our research objective. Therefore, two sentences were added to provide some insights into the accuracy of the RO temperature and RO wind record.

L93: "RO data have their highest precision and accuracy in the upper troposphere-lower stratosphere region, with uncertainty estimates of less than 0.7 K for individual temperature profiles from 8 km to 30 km and an accuracy of about 0.1 K (Scherllin-Pirscher et al., 2021). RO climate data records are therefore well suited for reliable trend analyses including benchmark quality temperature and geopotential height fields with a structural uncertainty of temperature trends of less than 0.05 to 0.1 K per decade for global to latitudinal means and a structural uncertainty of geopotential height trends of less than 4 m per decade within the RO core region (Steiner et al., 2013, 2020a)."

"As to the quality of the wind fields, values are within  $\pm 2$  ms-1 (Danzer et al., 2024; Nimac et al., 2025) and thus fulfil the WMO-OSCAR standard for horizontal wind information (see WMO-OSCAR, 2025, https://space.oscar.wmo.int/variables/view/wind\_horizontal, last access: 23 October 2025)."

**2: It would be interesting to include the tropical width trends from the RO data in Fig 5. How does that compare with the reanalyses for the same 15 years? How does that compare with previously published results if applicable (e.g, from Ao and Hajj 2012, Davis and Birner 2013)**

Thank you very much for your thought. However, the purpose of the figure is not to show trend estimates of the different data products, but rather to understand the needed time window for a reliable trend estimate. We emphasize, for example, in line 347, that we do not consider the currently studied RO record (14 years) long enough for trend studies of the tropical width.

For your convenience we include here the respective paragraph:

L347: "While we see great potential in RO records for the study of tropical atmospheric features, we also acknowledge their constraints. RO data has been successfully applied in climate studies, for instance, to investigate atmospheric temperature trends (e.g., Ladstädter et al., 2023; Steiner et al., 2020b) and tropopause trends (Ladstädter et al., 2025; Meng et al., 2021) over recent decades. However, for trend studies of the tropical width, which exhibits large variability, the record length is not yet considered long enough using regionally resolved RO data since 2006."

**3: Since the authors have 2D gridded dataset, can we say something about the regional trends? Is it possible that some regions show greater trends detectable over shorter periods?**

We consider these questions as a very interesting topic. While currently the RO record is too short for trend detection of zonal mean metrics, trends might arise also on shorter time periods when studying changes for individual regions. Nevertheless, we have to emphasize that trends were not the purpose of the underlying study. The purpose (for now) was to demonstrate the potential that lies within RO data to capture tropical width metrics. In the future, the Wegener Center will release a new data set with an extended record period of 22 years (2002-2024). We intend to conduct a follow-up study on this specific topic with the extended data. Furthermore, we added a sentence on this subsequent research interest:

L353: "A revenue of subsequent work will be to focus on regional trends, which may manifest over shorter timescales due to a more pronounced climate signal in certain geographic areas (e.g., Manney and Hegglin, 2018; Martin et al., 2020)."

**#4: What can we say/do with the different (indirect) metrics giving different trends?**

Although these metrics (i.e., tropopause break, eddy driven jet, subtropical jet) have been bundled together in the term "tropical width metrics" the features they describe and how these are changing aren't necessarily correlated. This has been shown by recent literature as outlined in the introduction of the manuscript (lines 54-60). Thus, also the physical message behind these different metrics does not necessarily coincide. While all metrics represent important features of the tropical atmosphere, with influence on the weather and climate of the subtropics, their specific impacts are more complex.

But we agree, we need to emphasize this more strongly in the manuscript and hence, added sentences in the introduction and method section.

L58: "All metrics represent important features of the tropical atmosphere, with individual influences on regional weather and climate patterns. Under climate change, shifts in these tropical width features can therefor lead to different changes in these subtropical patterns."

L112: "These metrics describe different features of the atmosphere at the tropical edge, a detailed description is given below:"

**5: Figure 4: some metrics show large zonal variation and systematic differences between RO and reanalyses over certain regions. Could you comment on implications on using the metric for monitoring tropical width changes, generally, or with RO?**

Thank you for this interesting question. We discussed the differences in the regional location of the metrics in chapter (3.2), we summarize here: With respect to the TPB the results align well between RO and reanalyses. Also, the STJ shows good results – all differences are minor with the exception of 180°W and 60°W in the northern hemisphere. Here it seems that the jet is shifted westward for RO. As for the EDJ, while in the southern hemisphere the metric seems to capture the location of the EDJ for RO in range of the one for reanalyses, in the northern hemisphere there are some notable differences. We assume that the systematic difference stems from the selection of the level. We conclude from our results that the temperature-based metrics can be used for monitoring purposes, consistent with ERA5 and JRA-3Q (MERRA-2 showed deviations over some regions). However, the wind-based metrics, specifically the EDJ in the northern hemisphere, needs to be viewed a little more carefully.

To express these differences and their relevance on the topic climate monitoring more clearly, we added some sentences in the results section 3.2:

L257: "Some exceptions occur over the NH, specifically for the EDJ metric there are some notable systematic differences over certain regions, which result most likely from the level selection of 850 hPa for reanalyses and 700 hPa for RO data (see also initial

validation in Fig. 1). Also, for the STJ in the NH, there are some longitudes at roughly 175 °W and 60 °W, where differences between the RO dataset and reanalyses occur."

We further extended and adapted a paragraph in the discussion as following:

L339: "We have proven the general applicability of RO data to locate jet streams, however, currently, the EDJ metric shows large systematic differences over some regions in the NH between RO and the reanalyses – this limits its utility for climate monitoring. Furthermore, other methods to define the jets exist (Keel et al., 2024; Liu et al., 2021), which were not tested in the process. Hence, a detailed investigation of further jet metrics using RO observations presents an interesting direction for future research to gain more insight into the jet response to climate change."

**6: Abstract: "The tropics are expanding poleward as a result of anthropogenic climate change." I think it's more accurate to say that "Many studies have shown that the tropics are expanding..."**

Thank you for pointing this out. Since this comment coincides with a comment from the other reviewer, we combine here the answer and rephrased the first sentence in the following way:

L7: "The tropical width is changing, with a poleward expansion being linked to anthropogenic climate change."

**References**

[revised manuscript text omitted]